# Study protocol for an exploratory interventional study investigating the feasibility of video-based non-contact physiological monitoring in healthy volunteers by Mapping Of Lower Limb skIn pErfusion (MOLLIE)

Mirae Harford [1,2] Carlos Areia [1] Mauricio Villarroel,[2] Joao Jorge,[2] Eoin Finnegan,[2] Shaun Davidson,[2] Adam Mahdi,[2] Duncan Young,[1] Lionel Tarassenko,[2] Peter J Watkinson [1]

¹Critical Care Research Group, Nuffield Department of Clinical Neurosciences, University of Oxford, Oxford, UK
²Department of Engineering Science, Institute of Biomedical Engineering, University of Oxford, Oxford, UK

**Correspondence to**
Dr Mirae Harford;
mirae.harford@ndcn.ox.ac.uk

## ABSTRACT

**Introduction** Skin perfusion varies in response to changes in the circulatory status. Blood flow to skin is reduced during haemodynamic collapse secondary to peripheral vasoconstriction, whereas increased skin perfusion is frequently observed when haemodynamics improve with resuscitation. These changes in perfusion may be monitored using non-contact image-based methods. Previous camera-derived physiological measurements have focused on accurate vital signs monitoring and extraction of physiological signals from environmental noise. One of the biggest challenges of camera-derived monitoring is artefacts from motion, which limits our understanding of what parameters may be derived from skin. In this study, we use phenylephrine and glyceryl trinitrate (GTN) to cause vasoconstriction and vasodilation in stationary healthy volunteers to describe directional changes in skin perfusion pattern.

**Methods and analysis** We aim to recruit 30 healthy volunteers who will undergo protocolised infusions of phenylephrine and GTN, followed by the monitored and timed release of a thigh tourniquet. The experimental timeline will be identical for all participants. Measurements of traditionally used haemodynamic markers (heart rate, blood pressure and stroke volume) and camera-derived measurements will be taken concurrently throughout the experimental period. The parameters of interest from the image data are skin colour and pattern, skin surface temperature, pulsatile signal detected at the skin surface and skin perfusion index.

**Ethics and dissemination** This study was reviewed and approved by the Oxford University Research and Ethics Committee and Clinical Trials and Research Governance teams (R63796/RE001). The results of this study will be presented at scientific conferences and published in peer-reviewed journals.

**Trial registration number** ISRCTN10417167.

## Strengths and limitations of this study

► Our novel study design within the field of optical monitoring allows a description of the chronological order and pattern of leg perfusion at the skin surface level.

► Using pharmacological challenges will allow us to answer definitively whether physiological signals of interest are present at skin level without artefacts created by movement.

► There is strong physiological and anatomical reasoning to believe that skin perfusion can be a sensitive marker of cardiovascular collapse compared with other traditionally used haemodynamic markers such as urine output or Glasgow Coma Scale.

► This study is limited by the use of healthy subjects to determine the normative patterns in the parameters of interest, rather than monitoring the ultimate population of interest (critically unwell).

## INTRODUCTION

Clinical examination adds valuable information to the overall haemodynamic assessment of a critically unwell patient.[1] The pattern of distribution of blood to the body can assist the assessment of the global circulation.[2] As an organ, the perfusion of the skin can indirectly portray the functional status of the cardiovascular system,[1 3 4] akin to the use of urine output to demonstrate adequate renal perfusion during haemodynamic assessment. This is true for assessing the response to specific rescue therapies[5 6] as well as during the initial clinical assessment. As a less privileged site, the skin offers a unique target for non-invasive monitoring where deterioration may be picked up at an early point to allow

intervention. The ease of access to the skin also provides a rationale for exploring its use as a monitoring target.

Recent efforts in camera-derived non-contact monitoring have identified several parameters of interest that can be measured at the skin surface.[7] In the visible spectrum, image photoplethysmography (PPG) allows the measurement of beat-to-beat information, although the precise origin of this fluctuation measured at the skin surface is still not clear.[8] Thermal camera measurements in the long-wave infrared range traditionally targeted nasal temperature fluctuation to estimate the respiratory rate, but there have also been attempts to use a surface temperature gradient to estimate haemodynamic status.[9] Laser speckle contrast imaging (LSCI) is a technique that allows measurement of the flow of blood in the vessels immediately under the skin surface,[10] which can be achieved using laser within or outside the visible spectrum. LSCI's current use is mainly in imaging the cerebral blood flow, wound assessment and intraoperative vessel monitoring. Imaging within these different spectral ranges provide diverse information, as they will penetrate the skin surface to a variable degree.[11]

To date, efforts made in this field have used the clinically trained eye as the definitive model monitor. However, methods based on the function of our eyes are often limited to the visible wavelength spectrum. As our vision relies on the visible spectrum, this methodology allows minimal depth penetration. Other monitors used frequently in clinical practice use other spectral ranges. For example, oxygen carriage in the blood is measured by comparing the absorption of light corresponding to wavelengths of 660 nm (red) and 940 nm (infrared). Furthermore, the clinician uses multiple parameters to make an overall assessment of the haemodynamic status such as skin temperature, skin colour, peripheral pulses and clamminess.[12] Rather than relying on the accuracy and reliability of signal measured in one wavelength alone, the use of multiple parameters measured across a larger wavelength spectrum to reach a composite conclusion similar to our clinical practice may provide richer information about the circulatory state.

Movement artefact remains a key challenge in image-derived measurements,[9] even when the parameter of interest is technically relatively accessible (eg, heart rate or respiratory rate). In our previous study, we saw that exercise-induced 'physiological' cardiovascular changes created a significant degree of movement artefact that it was not possible to assess for certain whether skin carries a meaningful signal that could be detected during these periods.

This study aims to describe the dynamic skin response to induced changes in haemodynamics, using a range of non-contact cameras. Following the description of baseline perfusion pattern at the skin surface, we will then create controlled changes by giving escalating infusions of phenylephrine and glyceryl trinitrate (GTN) with the aim of constricting and dilating the surface vessels, respectively. Using these controlled pharmacological methods, we aim to induce variations in skin perfusion and assess the pattern change while the amount of blood under the skin changes in opposite directions. We hypothesise that the induced increase and decrease in skin perfusion (and the resultant changes in skin colour, surface temperature and blood flow velocity) may be detected non-invasively using cameras detecting signals within an outside the visible spectrum. The expected changes and the cameras being used to detect the changes are shown in table 1. In order to describe these changes with detailed knowledge of the lower limb vascular anatomy, we plan to use an arterial tourniquet and timed release to identify the arterial perforator vessel locations[13] that represent the dominant branch supplying each region of skin. All subjects will undergo the same study protocol without randomisation. This protocol follows the guidelines from the Standard Protocol Items: Recommendations for Interventional Trials (SPIRIT) 2013 Statement.[14]

## METHODS AND ANALYSIS
We aim to recruit 30 healthy subjects aged between 18 and 65 years, aiming for a balanced ratio of male to female participants. As this is an unexplored field, the effect size is not known and it is not possible to calculate sample size. Therefore, this number is based on a balance of adequate study size to describe intersubject differences and sufficient numbers of male/female participants and potential skin types being included, with practical considerations of data size for analysis and running of the study. It is expected that the majority of the healthy volunteers will be approached via word of mouth and posters. Details of

**Table 1** Expected changes with planned infusions

|  | Camera detecting changes | Expected changes with infusions | |
|---|---|---|---|
|  |  | Phenylephrine | Glyceryl trinitrate |
| Skin colour | Red/Green/Blue visible spectrum camera | Pale skin, reduced red tone with vessel constriction | Increase in red tone with vessel dilation |
| Skin temperature | Thermographic camera | Reduced temperature and thermal signal | Increased temperature and thermal signal |
| Skin perfusion | Laser speckle contrast imager | Reduced perfusion/flux | Increased perfusion/flux |
| Pulsatility | Red/green/blue visible spectrum camera | Reduced pulsatility in image plethysmography | Increased pulsatility in image plethysmography |

## Box 1 Inclusion and exclusion criteria

Inclusion criteria
► Healthy adults aged 18–65 years.
► Willing and able to given informed consent for participation in the study.
► Willing to remove hair on lower limbs by shaving or waxing 24 hours prior to the study visit.

Exclusion criteria
► Participants whose anatomy, condition or other required monitoring preclude the use of the camera equipment or thoracic bioimpedance monitor device. Examples include skin disorders such as eczema, scleroderma or psoriasis.
► Any degree of lower limb amputation.
► History of surgical intervention to the thigh/knee/lower leg, except for procedures not expected to permanently change blood flow pattern to the lower limb skin (eg, knee arthroscopy).
► Allergy to silver chloride ECG sensors.
► Hyperthyroidism (intravenous phenylephrine contraindicated).
► Any regular medication except oral contraception.
► Pregnant or breastfeeding.
► History or current neurological condition affecting peripheral circulation.
► History of cardiovascular disease where phenylephrine or glyceryl trinitrate are contraindicated.
► History of severe headaches.

study inclusion and exclusion criteria are shown in box 1. In order to improve the quality of the images taken and reduce the amount of noise in the images, participants will be asked to remove lower limb body hair at least 24 hours prior to the study visit. The study will take place at the Cardiovascular Clinical Research Facility within John Radcliffe Hospital, Oxford, UK where clinical devices and full resuscitation equipment are available.

### Experimental timeline
Each study visit will begin with informed consent taken by the lead clinician and a final eligibility check. A cardiovascular examination will be performed including an examination of the peripheries for signs of cardiovascular disease, palpation of central pulse and auscultation of heart sounds (to exclude any conditions making the pharmacological challenges unsuitable or unsafe). We will also measure participants' height, weight, Fitzpatrick skin scale and baseline vital signs observations including heart rate, blood pressure, respiratory rate and oxygen saturation. Participants with baseline systolic blood pressure above 140 mm Hg or baseline diastolic blood pressure above 90 mm Hg on repeat measurements will not be included in the study.

Following intravenous cannulation of the right upper limb, participants will be asked to remain supine without movement. Movement of the lower limbs will be limited by a foot rest limiting involuntary external rotation of the leg. All monitoring equipment (including reference monitors and camera equipment) will be attached to the participants and electronically time-synchronised.

The cameras will record the skin of the right lower limb only. Data from the cameras will be stored in encrypted portable data storage and transferred to secure university server as soon as possible.

The experimental period will begin with a 5 min rest with no interventions. After 5 min, we will administer phenylephrine hydrochloride intravenously over a 15 min period with an increase in dose every minute for the first 10 min. The planned dose escalation is described in the Drug infusion protocol subsection. Subjects will then be given a rest period of 20 min to allow washout of the phenylephrine. Following the washout period, we will administer GTN intravenously over a 15 min period with an increase in dose every minute for the first 10 min as per protocol. After another 20 min washout period, we will study the refilling pattern of the lower limb using a thigh tourniquet. We will use a Hemaclear (OHK Medical Devices, Haifa, Israel) to achieve exsanguination of the right lower limb and as a right thigh tourniquet. The tourniquet will remain in place for 1 min before being removed while we monitor the limb using the experimental and reference monitors.

Participants and researchers will not be blinded to the treatment being given.

Participation in the study is voluntary and it will be made clear to the participants during the informed consent process that they may withdraw from the study at any time during the study visit without citing a reason. In cases of early cessation of the study, the researchers will discuss with each participant whether they wish for the study recording to be deleted. Side effects experienced during the drug experiment will not automatically lead to the participant being withdrawn. After a recovery period of up to 30 min, participants may proceed to any remaining study parts if both the participant and the lead clinician are in agreement.

### Reference measurements
► Standard non-invasive blood pressure monitor using an inflating cuff, placed on the left upper limb.
► Thoracic bioimpedance monitor to estimate stroke volume. The ancillary data (ECG, impedance waveform and derived measures including acceleration index, velocity index, pre-ejection period and conventional pulse transit time) will also be recorded.
► A pulse transit time monitor, which integrates R-wave estimate, PPG and chest wall inductance monitoring.

### Camera equipment
► Red-Green-Blue (RGB) camera (Grasshopper3 GS3-U3-51S5C, FLIR Systems, Oregon, USA)—visible spectrum, three-channel, 60 fps (drug infusion periods) or 140 fps (tourniquet release).
► Commercial thermographic camera (A6750sc, FLIR Systems)—thermal spectrum, 60 ffps.
► Commercial LSCI (moorFLPI-2, Moor Instruments, Axminster, UK) with near-infrared laser diode 785 nm. Camera resolution 580×752 pixels, scan area 15×20

cm, 5 fps (drug infusion periods) or 25 fps (tourniquet release).

## Data monitoring

A dataset will be defined as 'full' when the following are available for the participant:

► Collection of background characteristics: height, weight, Fitzpatrick skin scale and baseline vital signs observations including heart rate, blood pressure, respiratory rate and oxygen saturation.

► RGB camera data to include periods of phenylephrine infusion, GTN infusion and timed tourniquet release.

► Commercial thermographic camera data to include periods of phenylephrine infusion, GTN infusion and timed tourniquet release.

► Commercial LSCI data to include periods of phenylephrine infusion, GTN infusion and timed tourniquet release.

► Reference measurements (blood pressure at 1 min interval, stroke volume, ECG, impedance waveform, acceleration index, velocity index, pre-ejection period and conventional pulse transit time, R-wave estimate, peripheral PPG and chest wall inductance monitoring) to include periods of phenylephrine infusion, GTN infusion and timed tourniquet release.

Any incomplete dataset will not count towards the target of 30 participants. The recording of the above dataset will be checked by the researchers during the study visit in real time and immediately poststudy visit prior to transfer to permanent storage.

Completeness of dataset and any unforeseen problems arising from the study will be discussed in weekly study management meetings between MH, CA, JJ and MV or in study supervision meetings between MH, DY, LT and PJW.

## Drug infusion protocol

Phenylephrine infusion will be prepared by mixing phenylephrine chloride with 0.9% sodium chloride solution to create a concentration of 100 µg/mL. The prepared solution will be drawn up into a 50 mL syringe. The infusion rate will be started at 0.2 µg/kg/min and increased by 0.2 µg/kg/min increments every 1 min. No further increases will be made once a 30% increase in mean arterial pressure is reached. The maximum infusion rate indexed to actual body weight will be set at 2 µg/kg/min and once this rate is reached, the infusion and all monitoring will continue for further 5 min. The maximum absolute initial infusion rate will be set at 14 µg/min, and the maximum absolute peak infusion rate will be 140 µg/min.

A pre-prepared GTN solution at a concentration of 1 mg/mL will be used for the study. The solution will be drawn up into a 20 mL syringe. The infusion will be started at 0.15 µg/kg/min and increased by 0.15 µg/kg/min every 1 min. No further increases will be made once a 30% fall in mean arterial pressure is reached. The maximum infusion rate indexed to actual body weight allowed will be set at 1.5 µg/kg/min and once this rate is reached the infusion and all monitoring will continue for

further 5 min. Both drugs will be delivered using an infusion pump connected to the intravenous cannula using a two-port connector. The maximum absolute initial infusion rate will be set at 10.5 µg/min, and the maximum absolute peak infusion rate will be 105 µg/min.

Any side effects reported by participants will lead to immediate reduction of the drug dose to the immediate preceding infusion rate and documentation of the effect on participant study record.

## Data collection and statistical analysis

The haemodynamic variables measured using reference devices will be continuously recorded and analysed offline. The LSCI monitor results will be collected using the manufacturer's own software (Windows-based control, image processing and analysis) and exported to MATLAB for analysis. All images collected via the RGB and thermal cameras will be saved onto the portable hard drive using purpose-built software and analysed offline.

The main outcome measures in this study are the data collected using video cameras. Three non-contact parameters will be measured from the three cameras and the skin perfusion pattern will be described using composite measures from three different cameras. From the RGB camera, the measurement of interest will be the remote PPG signal across the region of interest (ROI) and the gradient between high and low signal amplitude areas. From the thermal camera, measurements of radiance from ROI will be measured. From the LSCI, three ROI will be selected manually to be centred over the knee, middle of the thigh and lower leg. For each ROI, the average flux (manufacturer estimation of blood velocity) will be measured.

For each parameter taken from the cameras, we will compare the difference between the baseline (rest period prior to drug challenge) and the peak effect (final period of the drug challenge). All values taken throughout each phase of the experimental period will be plotted for a qualitative description of the effect of increasing each drug dose. The plots will be used to show the pattern of effect of vasoconstriction and vasodilation on the different parameters. We will use a paired t-test to test, at the significance level of 0.05, the null hypothesis that there is no difference between the means of baseline and peak effect of the drug challenge. If the assumption of the test is not met by the data, then we will use a nonparametric Wilcoxon-Mann-Whitney test instead.

Other parameters that will be measured from the camera images will include pulse transit estimation from PPG signal between proximal and distal ROI, the surface temperature gradient between proximal and distal ROI, proportion of visible skin surface with the pulsatile component, and colour changes with increasing phenylephrine and GTN infusion.

The timed tourniquet release will be monitored using the same triple camera set up. The outcome of interest from this phase is the regional pattern in reperfusion, with the particular aim of identifying the chronological

order of reperfusion of different skin regions. The reperfusion pattern will be described from colour and skin surface temperature changes, remote PPG appearance in segmented ROIs and flux changes. Any early filling areas identified will be labelled as potential arterial perforator vessel site and overlaid on images from the drug infusion phase of the study.

The data will be presented as mean±SD unless otherwise stated. P values <0.05 will be considered statistically significant.

## ETHICS AND DISSEMINATION

This study was reviewed and approved by the Oxford University Research and Ethics Committee and Clinical Trials and Research Governance teams (R63796/RE001). The study is registered with the ISRCTN registry. Participants will be sent a detailed participant information sheet and given at least 24 hours for consideration whether to partake in the study. The participant information sheet (online supplementary file 1) and the recruitment poster have been reviewed and approved by the ethics committee for a complete and fair description of taking part. We will obtain written informed consent from all participants using an approved consent form (online supplementary file 2). Any protocol amendments will be submitted to the Oxford University Research and Ethics Committee and Clinical Trials and Research Governance teams for approval before being incorporated.

The findings from the study may be disseminated using academic media including peer-reviewed journal articles, national and international conference presentations, social media (including Twitter), electronic mail within the University of Oxford and the internet including the departmental websites. The findings from the study will form part of a doctoral dissertation for MH. Authorship of journal articles related to this study will follow the ICMJE guidelines.[15] Due to the nature of the data being collected (video/image-based), it is impossible to completely anonymise the data source. This will be explicitly discussed during the informed consent. Participants will be identified only by a participant identification number on study documents and any electronic database. Documents that contain identifying data and/or information allowing linkage between participant identification number and personal data will be stored separately under strict access controls. The participants will retain the right to have their images deleted from storage at any time on request.

### Access to data

The data, stored in a proprietary format that is only readable by the review software, will be recorded onto an encrypted portable storage device that will be transferred by the researchers in secured storage to the University server. Except during transport, access to the data will be restricted to academic areas behind two limited access doors. All study documents will be stored securely and only accessible by study staff (Kadoorie Centre for Critical Care Research and Education and the Institute of Biomedical Engineering/Oxford University Centre of Excellence for Medical Engineering) and authorised personnel from the University of Oxford for monitoring and/or audit of the study to ensure compliance with regulations. All video data will be stored securely using industry-standard encryption methods. The video data form part of the research data and will be stored securely for 5 years after the release date of the last publication arising from the study.

### Ancillary and post-trial care

If a participant in this study is ever considered to have suffered harm through their participation in the study, then the University of Oxford as the sponsor has arrangements to provide compensation. Participants will be informed of he potential route for escalation of any issues that arise from the study via the study team or via the Chair of the Medical Sciences Interdivisional Research Ethics Committee (MS IDREC) at the University of Oxford as appropriate.

## DISCUSSION

To the best of our knowledge, this is the first study aiming to create a skin surface perfusion map of the lower limbs and study changes to the map with an increase and decrease in peripheral skin blood flow. This is one of a series of planned studies designed to answer how much information can be measured at accessible skin surface level using non-contact tools.

A limitation of this proposed study is that it is only including healthy volunteers in artificially controlled conditions with restricted movement. However, we feel that this is an important initial step as if no useful signal can be derived from the best possible environment with no movement artefacts, then further efforts at looking for this absent signal in more realistic situations may not be fruitful. Creating optimum conditions by limiting movement will allow us to isolate signals that are present and potentially allow a better understanding of these signals of interest so that they can be isolated more effectively within a noisy environment. Some concerns may be raised by the use of healthy population. The haemodynamic control requires a fine balance of our sympathetic and parasympathetic systems, and we know that this balance is not at baseline in the critically ill. However, the understanding of how the differing physiology in healthy and non-healthy populations translate to changes measured at the skin surface will expand our understanding of critical illness and haemodynamic collapse.

Another key challenge is that by using specific receptor agonists, we bypass the physiological pathway by which the skin blood flow is reduced or increased. In the healthy population used in the proposed study, we expect the vessel changes to cause subsequent changes in central cardiovascular balance. In critical illness, haemodynamic

collapse and low blood pressure lead to peripheral vaso-constriction and reduced peripheral circulation. Creating the same peripheral conditions using phenylephrine infusion will lead to a rise in blood pressure. However, our methodology will allow us to see the pattern of reduction and increase in skin perfusion. This will provide invaluable information when similar changes are tracked on the skin surface in patient populations.

**Contributors** MH conceived the study with guidance from PJW, DY, LT, MV and JJ. MH, CA, MV, JJ, EF and SD will conduct screening and data collection. MH and AM planned the statistical analysis. Analysis will be performed by MH, MV, JJ, EF and SD. MH prepared the first draft of this manuscript. All authors critiqued and edited the manuscript for intellectual content.

**Funding** This work was supported by NIHR Oxford Biomedical Research Centre (BRC) under Technology & Digital Health theme. NIHR Oxford BRC advocates open access. MH, CA and PJW are funded by NIHR Oxford BRC. The work of JJ was supported by the RCUK Digital Economy Programme, grant number EP/G036861/1 (Oxford Centre for Doctoral Training in Healthcare Innovation). JJ also acknowledges Fundacao para a Ciencia e Tecnologia, Portugal, doctoral grant SFRH/BD/85158/2012. MV was supported by the Oxford Centre of Excellence in Medical Engineering funded by the Wellcome Trust and EPSRC under grant number WT88877/Z/09/Z. Trial sponsor: University of Oxford, Oxford, UK.

**Disclaimer** The funders have had no role in the study protocol design or the preparation of this manuscript, and will have no role in the collection, management, analysis and interpretation of the data, or the writing of the final report.

**Competing interests** PJW is Chief Medical Officer for Sensyne Health and holds share options in the company. His organisation has received research funding from Sensyne Health not associated with this work. LT is a non-executive Director of Sensyne Health and holds share options in the company. LT is a non-executive Director of Oxehealth and holds shares in the company.

**Patient and public involvement** Patients and/or the public were involved in the design, or conduct, or reporting, or dissemination plans of this research. Refer to the 'Methods' section for further details.

**Patient consent for publication** Not required.

**ORCID iDs**
Mirae Harford http://orcid.org/0000-0003-2851-1577
Carlos Areia http://orcid.org/0000-0002-4668-7069
Peter J Watkinson http://orcid.org/0000-0003-1023-3927

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
