## [Reviewer comments · BMJ Open]

ARTICLE DETAILS

TITLE (PROVISIONAL)	A study protocol for an exploratory interventional study investigating the feasibility of video-based non-contact physiological monitoring in healthy volunteers by Mapping Of Lower Limb skin perfusion (MOLLIE)
AUTHORS	Harford, Mirae; Areia, Carlos; Villarroel, Mauricio; Jorge, Joao; Finnegan, Eoin; Davidson, Shaun; Mahdi, Adam; Young, Duncan; Tarassenko, Lionel; Watkinson, Peter

VERSION 1 – REVIEW

REVIEWER	Godai Kohei Graduate School of Medical and Dental Sciences, Kagoshima University, Japan
REVIEW RETURNED	26-Jan-2020

GENERAL COMMENTS	General comments The study is planned to “describe the dynamic skin response to induced changes in haemodynamics, using a range of non-contact cameras”. I think this study would be interesting because if we could monitor the patients’ vital signs without touching them, application of monitoring might expand. Although the protocol is well planned. I believe the authors need to add explanations in detail. Especially, the description of statistic methods is scarce. Major concerns 1. The authors state that "This study aims to describe the dynamic skin response to induced changes in haemodynamics, using a range of non-contact cameras" (Page 6, Line 3). The hypothesis is not clearly explained, and I do not understand the primary outcome of this study. If the authors are planning to analyze agreement of values measured by cameras with the reference values, please state so. The authors need to clearly state the study hypothesis.2. The authors state that they “use an appropriate statistical test”. What is the appropriate statistical test? I do not understand how the authors analyze the results. The authors need to “describe statistical methods with enough detail to enable a knowledgeable reader with access to the original data to verify the reported results” (the SAMPL Guidelines. Int J Nurs Stud. 2015; 52: 5-9). The SAMPL guidelines will help the author to describe statistical methods.3. I do not understand why the authors planned to analyze re-filling pattern using tourniquet. Please explain the reason.4. The authors state that “movement artefact remain a key challenge in image-derived measurements” (Page 5, Line 51). I
--

	think the reference thoracic bioimpedance monitor is also impaired by patients' movement. Please describe how the authors plan to reduce the artifacts. Minor comments  1. The instruction (https://bmjopen.bmj.com/pages/authors/) says that An Article Summary should contain up to five short bullet points, no longer than one sentence each. The last bullet point is longer than one sentence (Page 4, Line 18). 2. Please specify the "our previous study " (Page 5, Line 52). 3. What is the cardiovascular examination (Page 7, Line 55)? Please specify it. 4. Why the authors selected HemaClear (Page 8, Line 18)? The authors will be able to monitor skin with pneumatic tourniquets. HemaClear, however, will cover skin surface. The authors will not be able to monitor skin with HemaClear. 5. Please explain why the authors selected the 30 % increase/decrease of mean arterial pressure (Page 9, Line 47). 6. What is "This" in the sentence "This will be noted from colour" (Page 10, Line 45)? Please specify it. 7. I am not sure whether the protocol article needs discussion section (Page 12).
--	--

REVIEWER	Viktor Hamrefors Department of Clinical Sciences, Lund University, Malmö, Sweden.
REVIEW RETURNED	26-Jan-2020

GENERAL COMMENTS	A well designed and clearly presented study protocol. My only question is whether the authors have an intention of including a specific proportion of female/male subjects?
---

REVIEWER	Jan Bakker Erasmus MC University Medical Center Rotterdam The Netherlands
REVIEW RETURNED	07-Feb-2020

GENERAL COMMENTS	The authors embark on an interesting study. The protocol is well written. I have only two comments: Introduction: The reference to the work of Doyle and Taylor between brackets is a bit confusing. Maybe better to just discuss the link between peripheral perfusion abnormalities and these study outcomes. Methods: The authors infuse the vasoactive drugs in the RIGHT upper limb. However, they also measure cuff blood pressure in the same limb. This could result in intermittent obstructions of drug delivery to the system. Although this is not a manuscript problem but more a possible protocol problem the authors may want to reassess this element of the study protocol.
---

	Also related to the protocol (not the manuscript): Have the authors considered to use a model of regional perfusion. So arterial infusion in the upper/lower limb to induce vasoconstriction/dilation. This would allow the have much less systemic effects (where the authors now aim for a max 30% increase/reduction in blood pressure. The effects of these systemic effects on cardiac output/function will be very different and may thus interfere with the results. As the authors only want to assess the feasibility of video-based skin/limb perfusion and not study the effects of the drugs themselves the regional infusion could bear less risk while maintaining the possibility to test the device/method. In the exclusion criteria I think it would make sense to exclude volunteers having increased blood pressure at baseline. The current exclusion criteria use a history of cardiovascular disease but the volunteers might not (yet) have that history. I think especially important as they aim to recruit volunteers up to 65yr of age.
--	---

VERSION 1 – AUTHOR RESPONSE

Reviewer: 1

Reviewer Name: Godai Kohei

- The study is planned to “describe the dynamic skin response to induced changes in haemodynamics, using a range of non-contact cameras”. I think this study would be interesting because if we could monitor the patients’ vital signs without touching them, application of monitoring might expand. Although the protocol is well planned. I believe the authors need to add explanations in detail. Especially, the description of statistic methods is scarce.

Thank you for highlighting the value of non-contact monitoring and for the positive comments. We have expanded the planned statistical methods as described below.

- The authors state that "This study aims to describe the dynamic skin response to induced changes in haemodynamics, using a range of non-contact cameras" (Page 6, Line 3). The hypothesis is not clearly explained, and I do not understand the primary outcome of this study. If the authors are planning to analyze agreement of values measured by cameras with the reference values, please state so. The authors need to clearly state the study hypothesis.

Thank you for this comment and we agree that the hypothesis requires clearer explanation in the manuscript. We plan to increase (using glyceryl trinitrate) and decrease (using phenylephrine) the overall blood flow to the skin using pharmacological methods. This is to create similar conditions during increased and decreased blood flow to the skin in response to overall cardiovascular status (e.g. reduced skin perfusion during cardiovascular collapse from peripheral vasoconstriction). We hypothesise that the induced changes in skin perfusion will be detectable using non-contact camera-based monitors.

As blood flow decreases, the skin becomes more pale and we hypothesise that as well as colour change the pulsatile component detectable at skin surface using cameras (image photoplethysmography) will be reduced. The reduced perfusion also leads to cooler skin temperature which may be detected using thermal cameras. Finally, the reduced perfusion and flow may be detected using laser speckle contrast imager as reduced blood flow to the region. Opposite responses to these are expected as blood flow increases.

As is the case with clinical examination, we hypothesise that the change in perfusion created may be better detected by using a combination of several parameters rather than just one parameter in isolation (discussed in the introduction, paragraph 3). For example, pulsatile photoplethysmographic signal may be differentially detected depending on skin colour, but thermal changes would be robust

to different skin colours. Therefore, we have set our primary outcome measure as combined changes detected by the three camera monitors.

The hypothesised changes are summarised in a new table (Table 1).

- The authors state that they “use an appropriate statistical test”. What is the appropriate statistical test? I do not understand how the authors analyze the results. The authors need to “describe statistical methods with enough detail to enable a knowledgeable reader with access to the original data to verify the reported results” (the SAMPL Guidelines. *Int J Nurs Stud.* 2015; 52: 5-9). The SAMPL guidelines will help the author to describe statistical methods.

Thank you for this comment and referring us to SAMPL guidelines. As described in the main text, the primary comparison will be between baseline periods and peak infusion periods. Following the SAMPL guidelines on “reporting hypothesis test”, we now identify the variables in the analysis, state the hypothesis being tested, identify the name of the test, and report the alpha level.

As secondary analysis, we plan to qualitatively show how the increase in dose affect each of the parameters being assessed by plotting dose-value curve for remote/image photoplethysmographic signal, radiance (from thermographic camera), and average flux. The aim of the qualitative description is to show (if present) opposite effects of increasing dose of the two drug infusions.

We have clarified this within the third paragraph of “Data collection and statistical analysis”.

- I do not understand why the authors planned to analyze re-filling pattern using tourniquet. Please explain the reason.

The anatomy of the arterial blood supply to the skin is well understood. However, the time of arrival of a pulse wave on the skin surface does not necessarily correspond to the surface path length of the arterial tree to that point. This may be explained by the layout of the arterial tree. The blood supply to an area of skin originates from a dominant arterial branch named perforators that stem from the underlying main artery.

The identification of the location of perforators is needed in order to describe the pattern at baseline and interpret changes caused by increase and decrease in skin blood flow.

We have added the following text to the final paragraph of the introduction:

In order to describe these changes with detailed knowledge of the lower limb vascular anatomy, we plan to use an arterial tourniquet and timed release to identify the arterial perforator vessel locations [ref: Taylor GI, Corlett RJ, Dhar SC, et al. The anatomical (angiosome) and clinical territories of cutaneous perforating arteries: development of the concept and designing safe flaps. *Plast Reconstr Surg* 2011;127(4):1447-59.] that represent the dominant branch supplying each region of skin.

- The authors state that “movement artefact remain a key challenge in image-derived measurements” (Page 5, Line 51). I think the reference thoracic bioimpedance monitor is also impaired by patients’ movement. Please describe how the authors plan to reduce the artifacts.

Thank you for this comment. We have attempted to reduce motion artefacts by choosing a study design involving drug infusion rather than other commonly used methods of inducing cardiovascular changes involving movement (e.g. exercise). The volunteers will rest on an examination couch and asked to remain still, and involuntary leg movement will be limited by the use of a foot rest limiting involuntary external rotation of the leg. This has been further explained in paragraph 2 of section “Experimental timeline”.

- The instruction (<https://bmjopen.bmj.com/pages/authors/>) says that An Article Summary should contain up to five short bullet points, no longer than one sentence each. The last bullet point is longer than one sentence (Page 4, Line 18).

Thank you for this comment. We feel that the last bullet point discussing the main limitation of the study requires further explanation which was difficult to limit to one sentence. As this is further

explored in the “Discussion” section, we would be happy to limit this bullet point to “This study is limited by the use of healthy subjects to determine the normative patterns in the parameters of interest, rather than monitoring the ultimate population of interest (critically unwell)” if shortening this final bullet point is required.

- Please specify the “our previous study ” (Page 5, Line 52).

We have not cited this previous study as the published findings from this study (Harford et al., Journal of Medical Engineering & Technology, 2019 43:33-37) did not include the video data due to difficulty analysing the data due to motion artefacts.

- What is the cardiovascular examination (Page 7, Line 55)? Please specify it.

Thank you for this comment. This has been clarified in the text under section “Experimental timeline”. The section now reads “Each study visit will begin with informed consent taken by the lead clinician and a final eligibility check. A cardiovascular examination will be performed including an examination of the peripheries for signs of cardiovascular disease, palpation of central pulse, and auscultation of heart sounds (to exclude any conditions making the pharmacological challenges unsuitable or unsafe).”.

- Why the authors selected HemaClear (Page 8, Line 18)? The authors will be able to monitor skin with pneumatic tourniquets. HemaClear, however, will cover skin surface. The authors will not be able to monitor skin with HemaClear.

We required exsanguination of the limb prior to the tourniquet application in order to observe the chronological order of blood supply/vessel filling. During the study design process, we trialled the use of several types of tourniquet device. The use of pneumatic tourniquet required exsanguination of the limb using either an Esmarch bandage or a Rhys-Davies exsanguinator, both of which left linear marks on the skin surface affecting the post-tourniquet skin appearance. HemaClear did not leave any marks on skin and we visualised the skin surface by removing the stockinet left covering the leg as soon as the tourniquet was applied (similar to its intended use in lower limb surgery settings), leaving skin surface to be monitored using cameras.

- Please explain why the authors selected the 30 % increase/decrease of mean arterial pressure (Page 9, Line 47).

These limits were chosen as safety limits rather than aims for variation. As participation in the study is voluntary, we felt that allowing larger variation in blood pressure cannot be justified.

- What is “This” in the sentence “This will be noted from colour” (Page 10, Line 45)? Please specify it.

This has been revised to “The reperfusion pattern will be described from the colour and skin surface temperature changes, remote PPG appearance in segmented ROIs, and flux changes.”.

- I am not sure whether the protocol article needs discussion section (Page 12).

Thank you very much for this comment and we understand that a discussion section is not always required for a protocol article. We have chosen to include a discussion section to highlight the novelty of the study as well as discuss the limitations we are aware of and our reasons for choosing the methodology despite the limitations discussed. We would be happy to amend this section or remove it if the editorial team feel it is not necessary.

Reviewer: 2

Reviewer Name: Viktor Hamrefors

- A well designed and clearly presented study protocol. My only question is whether the authors have an intention of including a specific proportion of female/male subjects?

We aim to open recruitment to both male and female participants and will aim to recruit equal proportion. This may not be possible depending on the interest and participant availability. We have updated the first sentence of Methods and analysis section to “We aim to recruit 30 healthy subjects aged between 18 and 65 years, aiming for a balanced ratio of male to female participants.”.

Reviewer: 3

Reviewer Name: Jan Bakker

- The authors embark on an interesting study. The protocol is well written.
- Introduction: The reference to the work of Doyle and Taylor between brackets is a bit confusing. Maybe better to just discuss the link between peripheral perfusion abnormalities and these study outcomes.

Thank you for this comment. We have removed this sentence discussing the longer term effects of transient ischaemia in core organs.

- The authors infuse the vasoactive drugs in the RIGHT upper limb. However, they also measure cuff blood pressure in the same limb. This could result in intermittent obstructions of drug delivery to the system. Although this is not a manuscript problem but more a possible protocol problem the authors may want to reassess this element of the study protocol.

Thank you very much for this insightful comment. We have revised the design to apply the blood pressure cuff to the left upper limb and this has been corrected under the subsection “Reference measurements”.

- Also related to the protocol (not the manuscript): Have the authors considered to use a model of regional perfusion. So arterial infusion in the upper/lower limb to induce vasoconstriction/dilation. This would allow the have much less systemic effects (where the authors now aim for a max 30% increase/reduction in blood pressure. The effects of these systemic effects on cardiac output/function will be very different and may thus interfere with the results. As the authors only want to assess the feasibility of video-based skin/limb perfusion and not study the effects of the drugs themselves the regional infusion could bear less risk while maintaining the possibility to test the device/method.

Thank you for this insightful comment. We considered a regional perfusion model but decided against designing the study using this model for several reasons. In healthy volunteer population, we felt that arterial cannulation of either upper or lower limb to induce vasoconstriction or dilation may be too interventional leading to limited interest in participating in the study, as well as being associated with more complications compared to venous cannulation.

- In the exclusion criteria I think it would make sense to exclude volunteers having increased blood pressure at baseline. The current exclusion criteria use a history of cardiovascular disease but the volunteers might not (yet) have that history. I think especially important as they aim to recruit volunteers up to 65yr of age.

Thank you for this comment. We agree with excluding volunteers with increased baseline blood pressure. This is important as participants with undiagnosed and untreated hypertension may have vascular changes affecting the study data, as well as safety while administering phenylephrine and glyceryl trinitrate. This has been specifically added in the protocol under “Experimental timeline” section.

VERSION 2 – REVIEW

REVIEWER	Kohei Godai Graduate School of Medical and Dental Sciences, Kagoshima University, Japan
REVIEW RETURNED	28-Mar-2020

GENERAL COMMENTS	I believe that the authors adequately answered my concerns and the manuscript has improved significantly. I do not have any further comments.
---

REVIEWER	Viktor Hamrefors Department of Clinical Sciences, Lund University, Malmö, Sweden.
REVIEW RETURNED	24-Mar-2020

GENERAL COMMENTS	An interesting study protocol, the updated version has been improved.
---